# Alpinetin protects against iron overload related osteoarthritis via NRF2/HO-1 pathway

Dongling Cai[1], Zhaofeng Pan[2], Shaocong Li[2], Qi He[2], Baihao Chen[2], Miao Li[2], Jiacong Xiao[2], Fancheng Wan[2], Haibin Wang[3]*, Chi Zhou [3]*

1 Department of Orthopaedics, Panyu Hospital of Chinese Medicine, Guangzhou, China, 2 The School of Medicine, Guangzhou University of Chinese Medicine, Guangzhou, P.R.China, 3 Department of Orthopaedics, First Affiliated Hospital, Guangzhou University of Chinese Medicine, Guangzhou, China

☯ These authors contributed equally to this work.
*hipman@163.com (HW); zcmzy1@163.com (CZ)

**Data availability statement:** All relevant data are within the paper and its Supporting Information files.

**Funding:** This study was supported by the National Natural Science Foundation of China under Grant Number [82074462]; Guangzhou University of Traditional Chinese Medicine

## Abstract

### Context

Alpinetin(APT) is a natural product with anti-inflammatory and antioxidant effects. Iron overload has been recognized in recent years as a new way to exacerbate osteoarthritis.

### Objective

This study evaluated the effects of ATP on iron overload related osteoarthritis.

### Materials and methods

C57BL/6J mice were randomly allocated to five groups as follows (n = 10 mice each): (1) sham; (2) destabilized medial meniscus(DMM); (3) DMM + ID; (4) DMM + ID + APT-L (50 mg/kg APT gavage daily); (5) DMM + ID + APT-H (100 mg/kg APT gavage daily). The chondrocytes treated by FAC (100μM) were used as an in vitro model of iron overload and the effect of APT was observed. Flow cytometry, fluorescence microscopy, Western blot, qRT-PCR and micro-CT were used to detect the mechanism of action of the APT.

### Result

Our studies showed that APT improved the viability of chondrocytes induced by iron overload. APT can reduce apoptosis of chondrocytes (19.41 ± 2.12% vs. 9.82 ± 1.74%). Furthermore, APT was found significantly attenuated ROS accumulation (2.04 ± 0.31 vs. 1.44 ± 0.15-fold) of chondrocytes through upregulating antioxidant genes NRF2 (1.18 ± 0.13 vs. 1.55 ± 0.17-fold) and HO-1 (1.27 ± 0.15 vs. 1.77 ± 0.20-fold). In vivo experiments revealed that APT attenuated cartilage damage (OARSI score 5.75 ± 1.32 vs. 3.75 ± 0.96) and subchondral bone proliferation in iron overload osteoarthritis mice.

"double first-class" and discipline coordination of high-level universities innovative team project [2021xk53].: The funders had no role in study design, data collection and analysis, decision to publish, or preparation of the manuscript.

**Competing interests:** The authors have declared that no competing interests exist.

## Conclusions

Our results show that APT can attenuate iron overload-induced cartilage damage *in vivo* and *in vitro* via the NRF2/HO-1 pathway. We demonstrated for the first time that APT has promising applications in iron overload diseases.

---

## Introduction

Osteoarthritis (OA), is a kind of disease that affects almost 2.4 billion people [1]. Any change that alters the biochemistry/metabolism of cells such as obesity, aging, gender, or genetics, is a risk factor for OA [2]. With the emergence of an increasing number of elderly and overweight individuals in society, the prevalence of OA has also increased yearly [3]. Generally, OA often affects the knees, causing pain, constraining patients body movement, and decreasing their quality of life [4]. Moreover, owing to the lack of effective treatment methods for OA, most patients suffer from pain, limited mobility, and other problems that eventually lead to disability or require joint replacement surgery. Therefore, new drugs that can mitigate the harmful effects of knee osteoarthritis (KOA) need to be developed to alleviate or treat KOA.

Pathologically, the main manifestations of KOA include synovitis, cartilage destruction, sclerosis of the subchondral bone with osteophyte formation, inflammation, and fibrosis of the infrapatellar fat pad [5,6]. Iron overload is broadly recognized as an essential factor to the injury [7–9]. An individual's body can accumulate excessive amounts of iron owing to aging, blood transfusions, diet, and genetic mutations. On the one hand, excessive accumulation of iron will damage the physiological functions of human tissue cells, not only reducing mitochondrial membrane potential, but also causing ferroptosis of human cells. On the other hand, pro-inflammatory cytokines in the OA environment disrupt iron homeostasis in chondrocytes by upregulating the expression of the iron influx mediator TfR1 and downregulating the iron efflux mediator FPN, exacerbating iron-induced cartilage injury. Eventually, accompanied by functional impairment due to increased iron overload, damage progression in various diseases such as KOA will continue to intensify [10].

Abnormal iron metabolism is associated with disease progression in OA, and excess iron adversely affects chondrocytes in OA. However, the mechanism of action of iron in KOA has not been fully elucidated, and pharmacological treatments need to be improved. Increasing evidence suggests that iron overload will increase intracellular reactive oxygen species (ROS) levels, causing cellular oxidative damage. Nuclear factor erythroid 2-related factor 2 (NRF2) and heme oxygenase 1 (HO-1) are key antioxidant enzymes that induce and regulate intracellular iron metabolism and concentration to prevent the oxidative damage caused by iron overload [11]. Alpinetin (APT) is a flavonoid that is abundant in the Chinese herb *Curcuma longa L.* (Zingiberaceae). Previous studies have recently reported that APT has an anti-inflammatory effect on the human body and a protective effect against cartilage tissue damage [12,13]. Additionally, APT could effectively upregulate NRF2 and HO-1 dose-dependently to alleviate oxidative damage [14].

In this study, experiments on an iron overload-induced KOA model were conducted to explore the effects and underlying mechanisms of APT. APT may alleviate iron overload-induced KOA by reducing peroxidative damage to cells. Furthermore, *in vitro* experiments were conducted on related key genes and signaling pathways, and the efficacy of the drug was verified in mouse experiments. The effects of APT on cell viability, apoptosis, and ROS production were analyzed. This study explores the impact of APT on cell viability, apoptosis, and ROS production, offering a theoretical foundation for its clinical use in treating iron overload-induced KOA.

## Materials and methods

### Materials and reagents

Penicillin/streptomycin (P/S), fetal bovine serum (FBS) (Scoresby, Australia), trypsin, and TRIzol Reagent were sourced from Thermo Fisher Scientific (USA). DMEM/F12 medium, phosphate-buffered saline (PBS), and 1% toluidine blue staining solution were provided by Servicebio Technology (China). Ferric ammonium citrate (FAC), iron dextran, and collagenase were purchased from Sigma-Aldrich (USA). Deset Biotech supplied the APT (Fig 1A, CAS: 36052-37-6, HPLC ≥ 98%). A cell counting kit (CCK8) was obtained from GlpBio Technology (China). RIPA buffer, a bicinchoninic acid (BCA) detection kit, a SDS-PAGE gel preparation kit, and 2′,7′-dichlorofluorescein diacetate (DCFH-DA) were from Beyotime Biotechnology (China). Accurate Biology was performed using the SYBR Green PCR Master Mix and the Evo M-MLV reverse transcriptase (RT) kit. The ECL chemiluminescent substrate kit was purchased from Biosharp Biotechnology (China). The primary antibody was purchased from Santa Cruz Biotechnology (USA). ML385 were purchased from MedChem Express (USA).

### Cell isolation and culture

The knee cartilage from 3-week-old male C57BL/6J mice (purchased from the Guangdong medical laboratory animal center, SCXK (YUE) 2018–0034) was isolated by cervical dislocation and minced into approximately 1-mm³ pieces. The tissue was digested with 0.25% trypsin and 0.25% collagenase II for 25 min and 6 h respectively at 37°C, and then the cells were collected and cultured in an atmosphere of 5% $CO_2$ at 37°C with DMEM/F12 medium (Servicebio, China) containing 10% FBS (Gibco, USA). After passaging, when the cells reached 80% confluence, the chondrocytes at passage 2 were used.

### CCK8 cell viability assay

Cell counting kit-8 (CCK8, GlpBio, USA) was used to evaluate the viability of chondrocytes. 96-well plates were seeded with chondrocytes at a density of $4 \times 10^3$ cells per well and cultured in DMEM/F12 medium (Servicebio, China) supplemented with 10% FBS (Gibco, USA) and 1% P/S (Gibco, USA). The viability of chondrocytes was examined by intervening with different doses of APT (0, 2.5, 5, 10, 40 and 80 µM, respectively) of APT (DeSiTe Biological Technology, China) for 48 h at 37°C. Each well was then incubated overnight at 37 °C, lucifugally, with 10 µL CCK8 solution. Finally, a microplate reader (Thermo Scientific, USA) was used to determine the absorbance at 450 nm. The optical density values of the different groups represented their viability. Upon detection of the appropriate concentration of APT, the cells were co-treated with 100 µM FAC to simulate an iron-overload environment. The cell culture method was the same as that mentioned above for APT (0, 5, 10, 20, 40 and 80 µM). The optimal therapeutic ATP concentration was determined repeatedly.

### Toluidine blue staining

The cells were then washed twice with PBS and fixed with 4% paraformaldehyde for 30 min at room temperature. The cells were then treated with a 1% toluidine blue staining solution (Servicebio) for 45 min at room temperature. After two washes with ultrapure water, the sections were rinsed with anhydrous alcohol (FUYU Chemical, China), air-dried, and observed under a microscope (Leica, Germany).

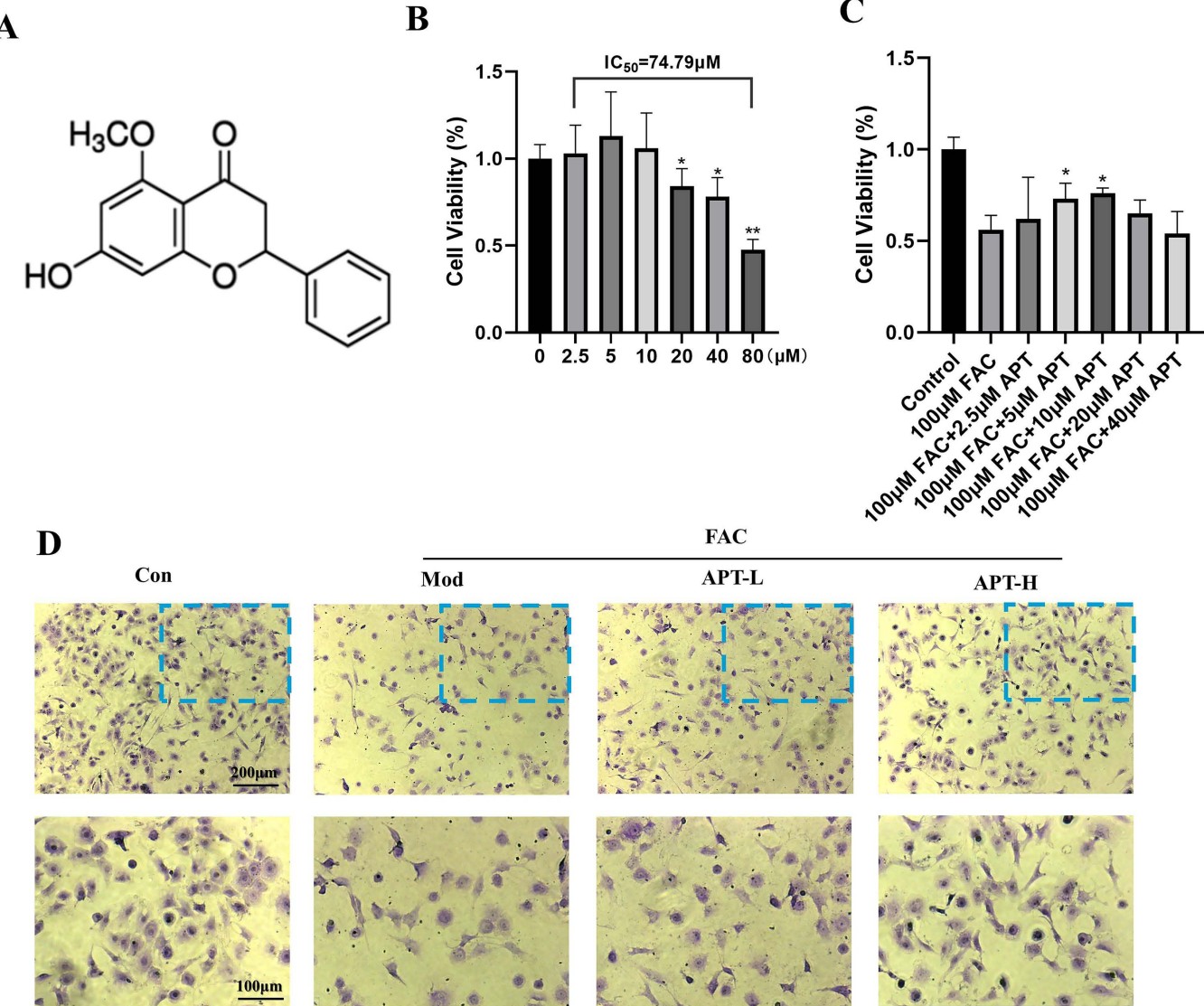

**Fig 1. Effect of APT on the viability of iron-overloaded chondrocytes or normal and the skill to synthesize type II collagen.** The chemical structure of APT (A). Changes in chondrocyte viability after APT intervention at concentrations of 0, 2.5, 5, 10, 20, 40, and 80 μM with or without 100 μM FAC (B, C). The ability to synthesize type II collagen was measured using toluidine blue (D), the darker the color the greater the ability. The second row of pictures shows the part of the first row of pictures surrounded by the blue dotted line.

### Reverse transcription and real-time polymerase chain reaction (RT-PCR) analysis

Chondrocytes were cultured in DMEM/F12 medium containing 10% FBS at a density of $3 \times 10^5$ per well. The medium was changed after 24 h, and the cells were divided to four groups as follows: Con group (cultured in DMEM/F12 containing 10% FBS); Mod group (cultured in DMEM/F12 with 10% FBS supplemented with 100 μM FAC); APT-L group (cultured in DMEM/F12 with 10% FBS, supplemented with 100 μM FAC and 5 μM APT); and APT-H group (cultured in DMEM/F12 with 10% FBS supplemented with 100 μM FAC and 10 μM APT).

To extract the total RNA from chondrocytes, the cells were incubated at 37 °C for 48 h, TRIzol Reagent (Invitrogen, USA) was used to lyse the cells. RNA quality was measured using a Nanodrop 2000 (Thermo Scientific, USA).

Complementary DNA (cDNA) was synthesized from total RNA using the Evo M-MLV RT Kit (Accurate Biology, China). The cDNA was then amplified with SYBR Green PCR Master Mix (Accurate Biology, China) under the following conditions: an initial polymerase activation at 95°C for 30 s, followed by 40 cycles of 95°C for 5 s, 60°C for 40 s, and a final extension step at 65°C for 5 s, with a 0.5°C increment for 60 s.

### Evaluation of apoptosis

An Annexin V-FITC/PI kit was used to detect apoptosis. Before being observed under the FACS-Canto™ II flow cytometer (BD Biosciences, USA), the cells were treated with drugs as mentioned above, washed thrice with PBS, and treated with Annexin V-FITC and PI for 20 min at 37 °C away from light. Washing the cells again after incubation is essential. The flow cytometry results were analyzed using the FlowJo software (ver. 10.6.2).

### Western blotting (WB) analysis

Before extract the chondrocytes using the RIPA buffer (Beyotime Biotechnology, China), the cells were washed twice with PBS. A BCA assay kit (Beyotime Biotechnology, China) was used to evaluate the total protein concentrations. Next, 8% or 10% sodium dodecyl sulfate-polyacrylamide gels (produced using the DS-PAGE Gel Preparation Kit, Beyotime Biotechnology, China) were prepared for electrophoresis. Protein samples (10 μg) were loaded and transferred onto polyvinylidene fluoride membranes (Millipore, USA). Then, the polyvinylidene difluoride membranes were blocked with 5% nonfat milk for 1.5 h at RT. Subsequently, the membranes were washed thrice and incubated with antibodies against Col II (1:1000, Affinity Biosciences, USA), matrix metalloproteinase 3 (MMP3) (1:1000, Affinity Biosciences, USA), Bax (1:1000, Affinity Biosciences, USA), NRF2 (1:1000, Affinity Biosciences, USA), HO-1 (1:1000, Affinity Biosciences, USA), and β-actin (1:2000, Affinity Biosciences) at 4°C overnight. The membranes were then incubated with a peroxidase-conjugated anti-rabbit IgG antibody (1:2000; Affinity Biosciences, USA) for approximately 1 h at room temperature. An ECL chemiluminescent substrate kit was used to visualize the chemiluminescent signals. Protein bands were semi-quantitatively analyzed using a gel imaging system (Bio-Rad). β-actin served as a loading control.

### Evaluation of intracellular ROS

The chondrocytes were incubated with DCFH-DA (Beyotime Biotechnology, China) in the dark for 25 minutes at 37°C to specifically measure intracellular ROS levels. After incubation, the cells were washed twice with PBS and observed under an inverted fluorescence microscope (Leica, Germany). Subsequently, the cells were resuspended in buffer and analyzed using a FACS Canto II flow cytometer (BD Biosciences, USA).

### Animal experiment

A total of 50 6-week-old C57BL/6J male mice (each weighing 20 ± 3 g from the experimental Animal Center of Guangzhou University of Chinese Medicine (SCXK (YUE) 2018–0034) were used in this study. The animals were randomly allocated to five groups as follows (n = 10 mice each): (1) sham group (surgery was performed without damaging the ligament); (2) destabilized medial meniscus (DMM) group (DMM surgery at 10 weeks of age; the remaining groups received DMM surgery at 10 weeks of age); (3) DMM + ID group (weekly 0.5 g/kg iron dextran was administered intraperitoneally at 8–16 weeks of age; this intervention was applied to the remaining groups); (4) DMM + ID + APT-L group (50 mg/kg APT was administered by gavage daily from 11–18 weeks of age); and (5) DMM + ID + APT-H group (100 mg/kg APT was administered by gavage daily from 11–18 weeks of age). All experimental animals were raised in hygienic plastic cages in SPF pathogen-free experimental animal facilities (Laboratory Animal Center, First Affiliated Hospital of Guangzhou University of Chinese Medicine, SYXK (Yue) 2018–0092. The mice were maintained on an alternating 12 h light/dark cycle with adequate water and food. All samples were harvested at 18 weeks of age, and complete knee samples were isolated for the next process(the mice were anesthetized with pentobarbital sodium 30 mg/kg). The experiments were approved by the

Review Board of the First Affiliated Hospital of Guangzhou University of Chinese Medicine (no. TCMF1–2021029; May 20, 2021).

## Micro-Computed Tomography (μ-CT) analysis

Complete knee samples were fixed in 4% paraformaldehyde. The samples were then wrapped in a non-woven fabric containing 75% ethanol (FUYU Chemical, China) for scanning using a Skyscan 1172 (Bruker, Belgium). The settings for the scanning mode were as follows: 80 kV, 100 μA, 0.4° rotation step, and 0.5 mm aluminum filter and 5-μm slice thickness. A custom analysis program (CTAn, SkyScan) was used to evaluate the data. The volume of interest mainly consisted of a 100-slice section from the medial to lateral side of the tibia.

## Histology assay

All samples were processed and fixed as described above, followed by decalcification in 14% ethylenediaminetetraacetic acid (EDTA) for 10 days. After dehydration, the skeletal samples were embedded in paraffin and sectioned into thin slices of 5 μm thickness. The slices were then stained with Safranin O-fast Green (S-O) and scanned using a Panoramic Midi digital slide scanner (3DHISTECH Ltd, Hungary).

## Statistical analysis

Each experiment was conducted at least three times, and the results are presented as mean ± standard deviation. Data were analyzed using a one-way analysis of variance, followed by post hoc comparisons between groups using Tukey's test in SPSS (version 25.0). Data visualization was performed using GraphPad Prism (version 8.0.2).

# Result

## Dose-dependent effect of APT on chondrocyte activity

In general, the CCK8 assay is used to assess the toxicity of APT in chondrocytes. Chondrocytes exhibited the greatest cellular viability when intervened with APT in 5–10 μM (Fig 1B). Moreover, iron overload significantly reduced cell viability, whereas APT reversed this iron overload-induced reduction (Fig 1C). Moreover, toluidine blue staining revealed the same effect as that of APT on chondrocytes. Both 5 and 10 μM APT reduced the damage to the collagen synthesis capacity of FAC-induced chondrocytes caused by FAC. The 10-μM concentration had better effects than the 5-μM concentration (Fig 1D).

## APT inhibited iron overload-induced chondrocytes apoptosis

To examine the protective effects of APT against iron overload-induced apoptosis, we used PCR and WB to detect the levels of apoptotic factors. COL II is the main component of the extracellular matrix of chondrocytes and is a functional indicator of their chondrogenic ability. MMP3 is a member of the matrix metalloproteinase family, which can digest matrix proteins and other extracellular matrix components, and is positively correlated with arthritis. FAC intervention significantly increased the levels of MMP3 and Bax in Mod group and decreased the levels of Col-II. APT demonstrated an antagonistic effect on FAC, increased the level of Col-II, and decreased the MMP3 and Bax expressions in APT-L and APT-H groups (Fig 2A–2E). Further studies using flow cytometry indicated that APT alleviated the abnormal increase in apoptosis caused by iron overload dose-dependently (Fig 2G).

## APT reduced iron overload-induced ROS accumulation

Excessive iron accumulation can trigger the Fenton reaction, leading to increased intracellular ROS and cytotoxicity [15]. Consistent with relevant research, a fluorescent probe of DCFH-DA was used for fluorescence imaging. The probe interacts with ROS, and the fluorescence intensity reflects the accumulation of ROS in cells. As previously reported, the

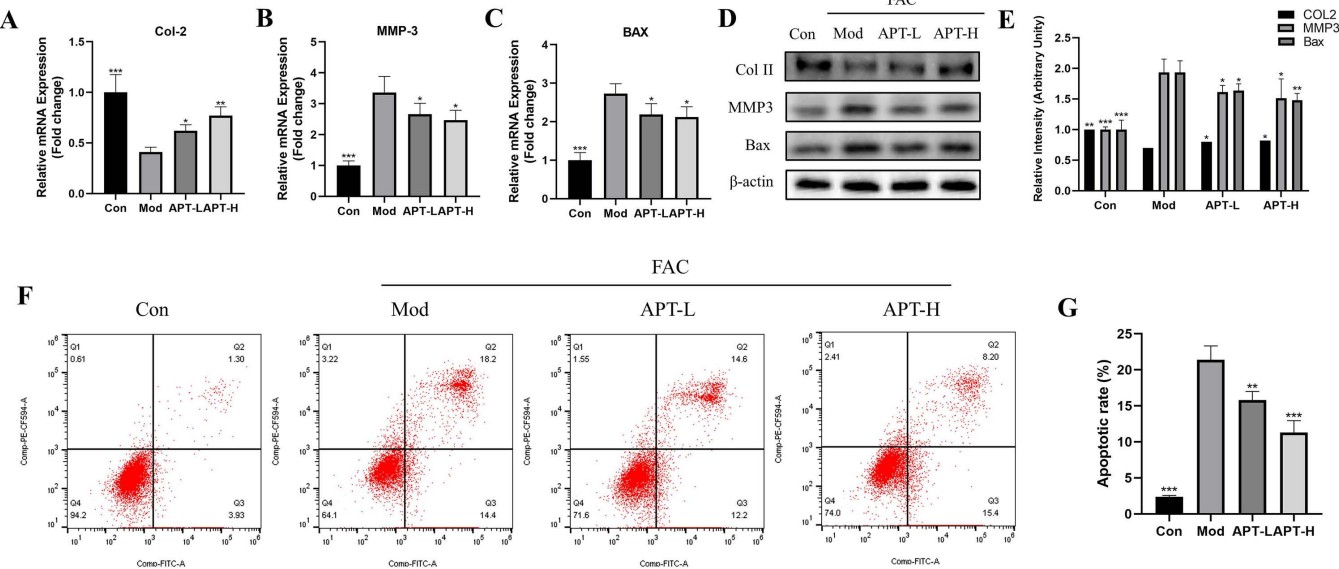

**Fig 2. APT contributes to the reduction of iron overload-induced chondrocyte arthritis and apoptosis.** Chondrocytes were treated with 100uM FAC and co-cultured with 0, 5, 10 μM APT to observe the expression levels of COL2 (A), MMP3 (B), and BAX (C) genes. Protein levels were further verified using western blot (D) and the statistical results are displayed in the bar graph (E). Further, we examined changes in chondrocyte apoptosis under the same culture conditions (F, G) and the results showed that APT reduced FAC-induced chondrocyte apoptosis in a concentration-dependent manner. (Data was manifested as mean±SD; *** P<0.001,** P<0.01, * P<0.05 vs Mod).

iron-overloaded state demonstrated stronger fluorescence in fluorescence microscopy, whereas the fluorescence intensity was reduced after APT intervention (Fig 3A and 3B). Furthermore, PCR and WB revealed that APT enhanced the NRF2 and HO-1 expressions in APT-L and APT-H groups (Fig 3C–3E).

## APT reduced ROS accumulation by regulating the NRF2-HO-1 pathway

These results suggest that APT affects ROS production through the NRF2-HO-1 pathway. To verify this hypothesis, WB was performed and under the action of the NRF2 inhibitor ML385 (5 μM), the HO-1 and Col II expressions were significantly decreased, whereas those of apoptosis factors MMP3 and Bax were significantly increased (Fig 4A and 4B). Flow cytometry and fluorescence staining revealed that intracellular ROS increased significantly after ML385 treatment in iron-overloaded chondrocytes, and APT effectively reversed this phenomenon. APT reduces the accumulation of intracellular ROS caused by iron overload via the NRF2-HO-1 pathway (Fig 4C–4F).

## APT attenuates iron overload-induced KOA lesions in mice

We also investigated whether APT could alleviate KOA progression *in vivo* by grouping mice and administering DMM and iron dextran intraperitoneally for KOA modeling. Cartilage wear in all the groups was compared to that in the sham group. Safranin O-fast green (S-O) staining indicated that compared to mice not injected with iron dextran, those injected with iron dextran demonstrated more evident thickness of the articular cartilage and surface defects, and the subchondral bone was also observed to have migrated up (Fig 5A). Severe cartilage wear and subchondral hypertrophy were observed in the DMM group. However, after intragastric administration of APT, both articular cartilage wear and surface defects improved compared to those in the DMM+ID group. As demonstrated by the osteoarthritis research society international (OARSI) score (Fig 5E), APT reduced the damage caused by iron glucan to the articular cartilage.

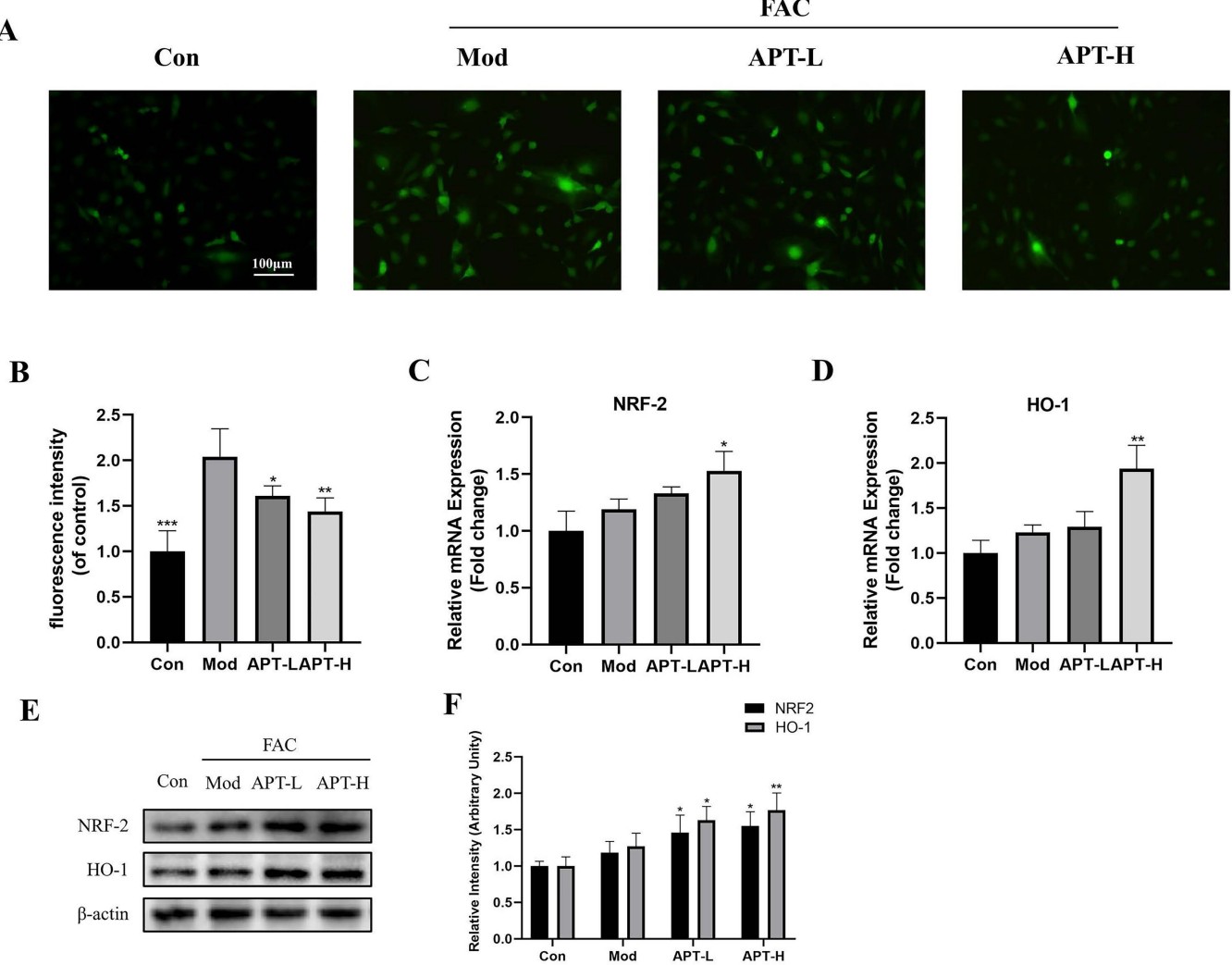

**Fig 3. APT reduces reactive oxygen species accumulation in chondrocytes following FAC intervention by activating antioxidant signaling pathways.** The intracellular ROS content was measured using DCFH-DA. Under fluorescence microscopy, the stronger the fluorescence intensity, the higher the level of ROS (A, B). The results suggest that APT attenuated the accumulation of ROS due to FAC. The gene expression (C, D) and protein content (E, F) of NRF2 and HO-1 were detected and their expression was found to be elevated by APT intervention. (Data was manifested as mean±SD; *** $P < 0.001$, ** $P < 0.01$, * $P < 0.05$ vs Mod).

To visualize the subchondral bone state in mice, 3-dimensional (3-D) modeling was performed using CTVOX (Fig 5A). Cross-sectional images demonstrated that after DMM treatment, evident bone hyperplasia appeared in the medial tibia of the mice in each group. Compared to the DMM group, the subchondral bone plate of the mice in the DMM＋ID group was relatively thinner, the trabecular bone was slender and disordered, and the subchondral bone was compressed and collapsed. Although APT treatment can alleviate abnormal changes in the bone trabecular structure, evident bone hyperplasia remains. Furthermore, 3-D CT and morphological analyses of the subchondral bone were performed to examine whether APT could alleviate disruption of the subchondral bone structure. Bone volume/tissue volume (BV/TV) was used to assess bone volume changes, trabecular separation/spacing (Tb. Sp) was used to calculate the intertrabecular distance, and connectivity density (Conn.D) was used to assess the density of trabecular meshwork connections [16]. BV/TV

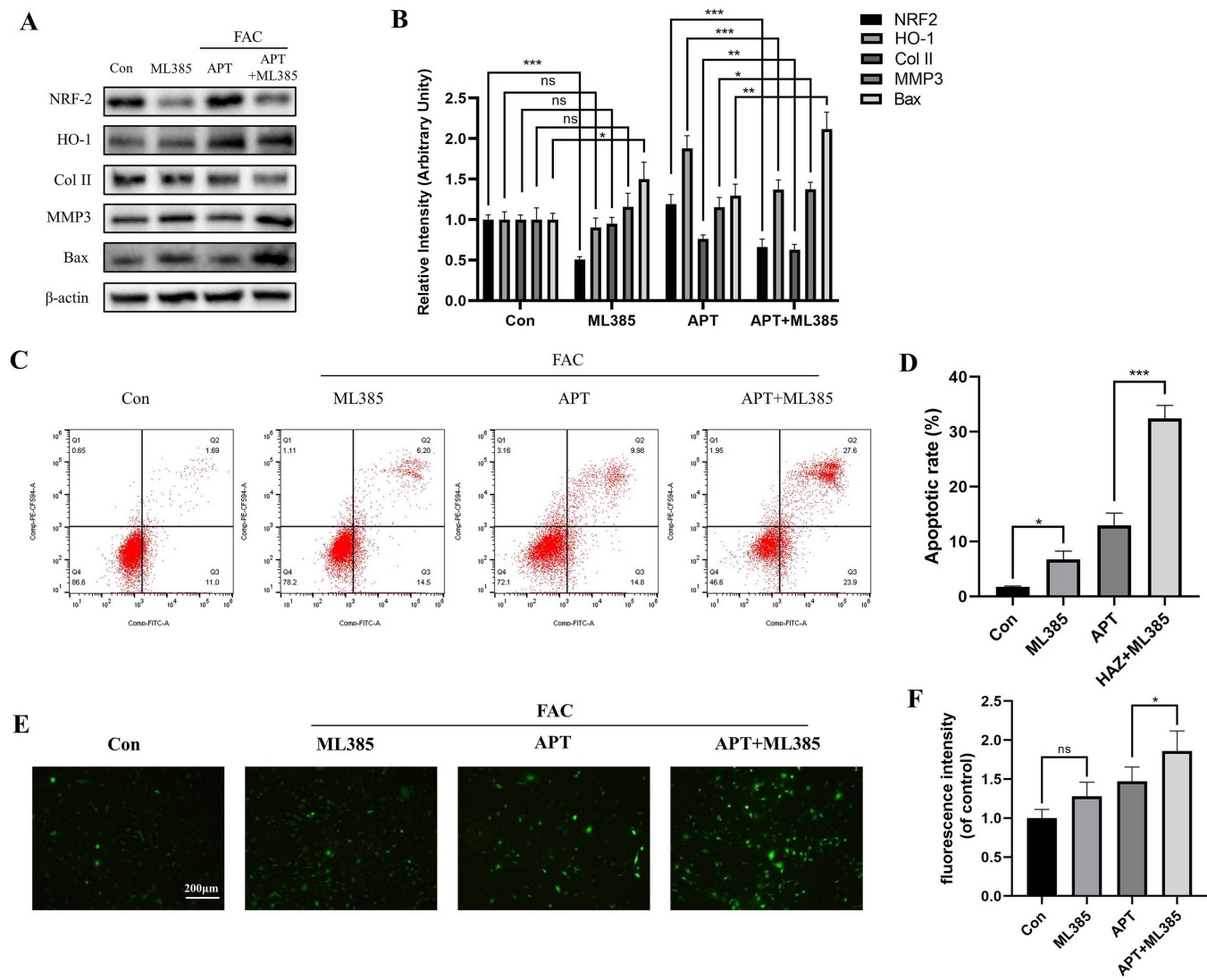

**Fig 4. ML385, a specific inhibitor of NRF2, reversed the protective effect of 10 µM APT on chondrocytes induced by iron overload.** The protein expression levels of COL II, MMP3, BAX, NRF2, and HO-1 were measured after the addition of ML385 (A, B). The apoptosis rate (C, D) and intracellular ROS accumulation (E, F) of chondrocytes were measured using flow cytometry under the same culture conditions. (Data was manifested as mean±SD; *** $P<0.001$,** $P<0.01$, * $P<0.05$ vs Mod).

was significantly higher in the DMM group than that in the DMM+ID group, whereas Tb. Sp decreased, suggesting more pronounced trabecular sclerosis. However, no significant difference in Conn D was observed between the DMM+ID and DMM groups (Fig 5B–5D).

After APT treatment, the BV/TV in the APT-L and ALT-H groups significantly increased compared to that in the DMM+ID group, and TB.Sp and Conn.D also decreased in the APT-H group. However, no significant differences in Sb. Sp and Conn.D were noted between the ALP-L and DMM+ID groups. This suggests that APT can slow the deterioration of subchondral bone microarchitecture in iron-induced joint damage, with higher concentrations having a more significant effect than lower concentrations, and has the potential to treat this disease.

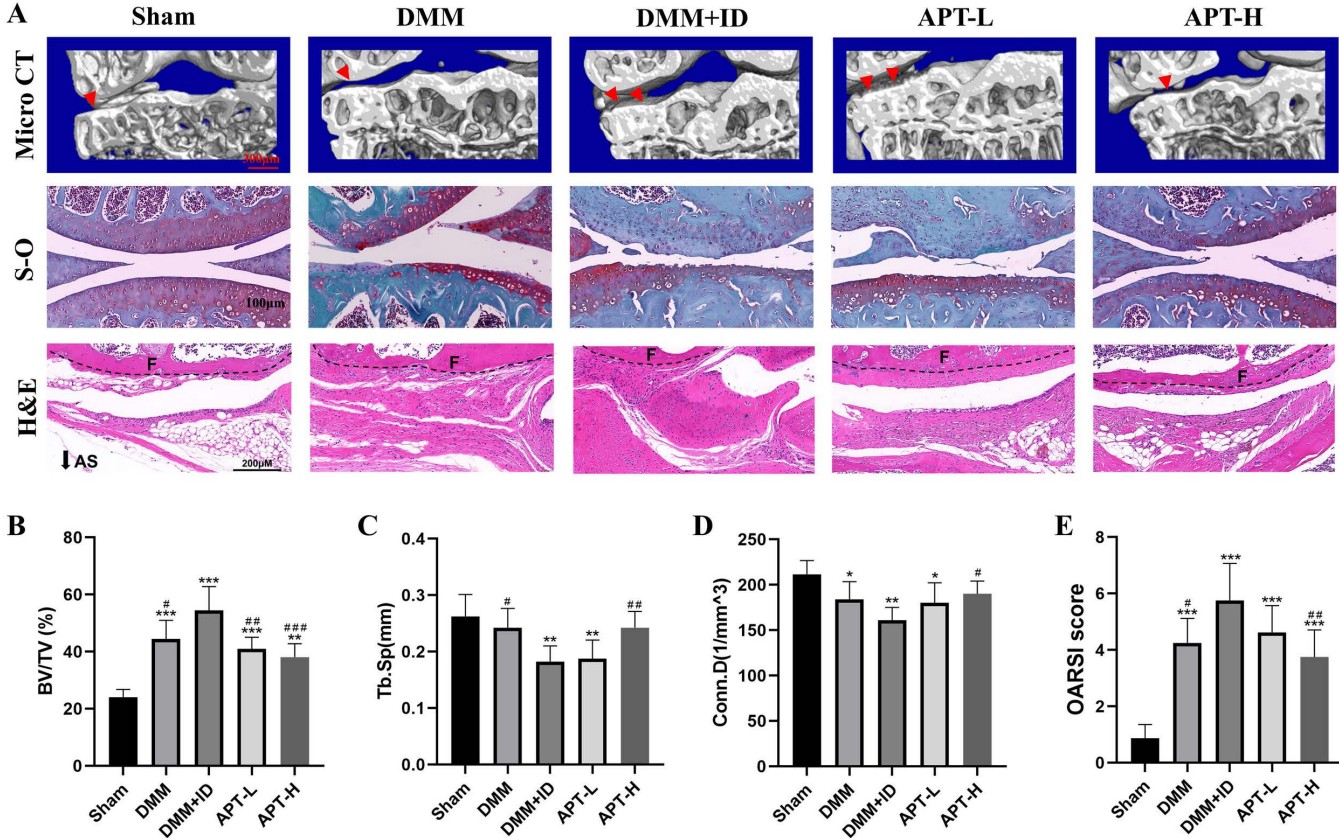

**Fig 5. APT attenuates iron overload and DMM procedure-induced cartilage damage and subchondral osteosclerosis *in vivo*.** An analysis of 3D reconstructions of micro-CTs.In the HE stained image, F represents the cortical bone of the femur(A). The red arrows indicate hyperplasia of subchondral bone. μ-CT can determine subchondral bone parameters such as the BV/TB, Tb. Sp, Conn.D were quantitatively analyzed (B, C, D). Cartilage damage was observed using a Safranin O-Fast Green staining (A). The OARSI score is presented as a bar graph (E).

## Discussion

The effect of iron overload on KOA has received increasing attention in recent years. Iron is an essential active metal in the body [17,18] and when the body's iron metabolism is disturbed or when diseases such as hemophilic arthritis occur, it tends to accumulate abnormally in the body [19,20] and causes further aggravation of KOA [21]. In addition to the need to limit exogenous blood transfusions to prevent excessive iron accumulation [22], treatment of primary or secondary disorders of iron metabolism is key to treating patients with iron overload in KOA; however, there is a considerable paucity of relevant therapies [23].

APT is a naturally occurring flavonoid widely used for its anti-inflammatory and biological activities [24–27]. Although the therapeutic effects of APT on KOA inflammation have been reported [13], its role in iron-associated KOA remains unclear. Our data indicate that APT attenuated cartilage degeneration and delayed subchondral osteosclerosis in mice with iron overload-associated KOA. Further experimental analyses are required to determine whether APT alleviates KOA or plays an additional role in iron overload-associated KOA.

Chondrogenic degeneration and abnormal chondrocyte death due to an imbalance between chondrocyte synthesis and catabolism are common manifestations in the pathogenesis of KOA. Iron overload can damage articular

cartilage either directly or by stimulating synovial inflammation [28]. FAC is an iron supplement commonly used to simulate the *in vitro* iron overload environment in different cells. In our study, the CCK8 and toluidine blue results from FAC-induced chondrocyte models of iron overload verified the inhibitory effect of iron overload on chondrocyte activity. APT may have a positive role in iron overload cartilage injury. The ECM of chondrocytes contains large amounts of Col II, which is the basis for chondrocyte secretion and formation of a normal cartilage matrix. The ECM of chondrocytes is normally degraded by matrix metalloproteinases such as MMP3 and MMP13. However, under normal conditions, the expression of matrix metalloproteinases is severely restricted and is only activated in an inflammatory environment while mediating the apoptosis of chondrocytes, which manifests as cartilage degeneration *in vivo*. In our study, WB and qRT-PCR results indicated that APT effectively mitigated the impact of iron overload by promoting the expression of MMP3, MMP13, and Bax, and inhibiting the expression of Col II. This suggests that APT protects iron-overloaded chondrocytes and helps maintain ECM stability. Flow cytometry revealed that APT inhibited chondrocyte apoptosis caused by iron overload. Thus, APT reduces the impaired function of active inhibitors in chondrocytes under iron overload.

Iron overload can lead to the excessive accumulation of ROS, which in turn promotes apoptosis in chondrocytes. Intracellular iron overload promotes ROS production through the Fenton reaction, which can easily induce oxidative damage when cells are unable to immediately clear excess ROS [29]. This damage may induce apoptosis [30,31]. Using the fluorescent probe DCFH-DA, we observed that iron overload significantly increased intracellular ROS levels, and that APT alleviated this process. However, the mechanism by which APT inhibits the increase in intracellular ROS levels caused by iron overload requires further exploration.

NRF2 in mammals is responsible for the regulation of cellular redox homeostasis and protective antioxidant activity in mammals [32]. HO-1, a downstream gene regulated by NRF2, is one of two isoforms in mammals. As an inducible 32-kDa protein, it is highly upregulated by many stimuli, including heme, nitric oxide, heavy metals, growth factors, cytokines, and modified lipids [33]. Some studies have reported NRF2-HO-1 as the main cellular antioxidant system and a key factor in mitigating iron overload damage [2]. In our study, when chondrocytes were treated with FAC, the expression of both NRF2 and HO-1 was upregulated. This suggests that the NRF2-HO-1 pathway is activated to exert antioxidant effects when the body is overloaded with iron. Interestingly, the NRF2 and HO-1 expressions were further upregulated after the treatment of cells with APT. This suggests that APT activates the antioxidant system and counteracts ROS accumulation in iron-overloaded environments. To verify whether the protective effect of APT on cell function was mediated by the NRF2-HO-1 system, we used ML385 to inhibit the activation of NRF2. The results revealed that ML385 inhibited the effect of APT by increasing the HO-1 expression in an iron-overloaded environment and increasing the expression of the apoptotic genes MMP3 and BAX. The reduction in NRF2 expression was accompanied by an increase in intracellular ROS content, and the effect of APT on reducing ROS production was substantially attenuated by ML385. Furthermore, flow cytometry indicated that the apoptosis-inhibiting effect of APT was reversed by ML385. In conclusion, APT limits iron overload-induced chondrocyte injury via the NRF2-HO-1 pathway. We believe that APT has the potential for further research in the treatment of iron overload-related arthritis.

Our study had some limitations. First, mouse chondrocytes remain different from human chondrocytes to some degree, and the chondrocytes selected from 3-week-old mice were also in a proliferative state, with results that may differ from those of adult chondrocytes. Second, this study focused on cartilage destruction and changes in subchondral bone and did not address the effects of the infrapatellar fat pad, meniscus, and synovium, despite their important roles in KOA. Finally, other flavonoids that activate the NRF-2/HO-1 pathway may also be used to treat iron overload-related KOA but were not explored in the present study. The results of this study suggest that APT protects iron overload-associated chondrocytes and maintains matrix stability through activation of the NRF2-HO-1 pathway. In addition, exploring the protective effects of other flavonoids on iron overload OA is a possible idea to find a more optimal treatment for OA and improve patients quality of life by comparing the efficacy of different flavonoids.

## Conclusion

This study reveals an important role for APT limits iron overload-induced chondrocyte injury via the NRF2-HO-1 pathway. We believe that APT has the potential for further research in the treatment of iron overload-related arthritis.

## Supporting information

**S1 Fig. Adult mouse chondrocytes undergo apoptosis in an iron-dependent manner.** Chondrocyte apoptosis assay in adult mice (A,B); P2 Alisin blue staining of adult mouse chondrocytes (C).
(TIF)

**S2 Fig. PCR results of DMT1, TFR1, and FPN with cellular iron accumulation.** PCR results of DMT1 (A); PCR results of TFR1 (B); PCR results of FPN(C); Fluorogram and fluorescence quantification of calcineurin (D,E)。
(TIF)

## Author contributions

**Writing – original draft:** Chi Zhou, Dongling Cai, Zhaofeng Pan, Shaocong Li, Qi He, Baihao Chen, Miao Li, Jiacong Xiao, Fancheng Wang, Haibin Wang.

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
