## [Decision Letter · Decision Letter 0]

PONE-D-24-39558
Alpinetin protects against iron overload related osteoarthritis via NRF2/HO-1 pathway
PLOS ONE

Dear Dr. Zhou,

Thank you for submitting your manuscript to PLOS ONE. After careful consideration, we feel that it has merit but does not fully meet PLOS ONE’s publication criteria as it currently stands. Therefore, we invite you to submit a revised version of the manuscript that addresses the points raised during the review process.

We look forward to receiving your revised manuscript.

Kind regards,

Xindie Zhou

Academic Editor

PLOS ONE

Naringenin protects against iron overload-induced osteoarthritis by suppressing oxidative stress - https://doi.org/10.1016/j.phymed.2022.154330

Cardamonin protects against iron overload induced arthritis by attenuating ROS production and NLRP3 infammasome activation via the SIRT1/p38MAPK signaling pathway - https://doi.org/10.1038/s41598-023-40930-y

(Among others)

In your revision ensure you cite all your sources (including your own works), and quote or rephrase any duplicated text outside the methods section. Further consideration is dependent on these concerns being addressed.

“This study was supported by the National Natural Science Foundation of China under Grant Number [82074462]; Guangzhou University of Traditional Chinese Medicine "double first-class" and discipline coordination of high-level universities innovative team project [2021xk53].”

6. We note that your Data Availability Statement is currently as follows: [All relevant data are within the manuscript and its Supporting Information files.]

Reviewers' comments:

Reviewer's Responses to Questions

**Comments to the Author**

1. Is the manuscript technically sound, and do the data support the conclusions?

Reviewer #1: Yes

2. Has the statistical analysis been performed appropriately and rigorously? 

Reviewer #1: Yes

3. Have the authors made all data underlying the findings in their manuscript fully available?

Reviewer #1: Yes

4. Is the manuscript presented in an intelligible fashion and written in standard English?

Reviewer #1: Yes

5. Review Comments to the Author

Reviewer #1: This manuscript investigates the effects of alpinetin on iron overload-related osteoarthritis. Using both in vivo and in vitro models, the study demonstrates that APT reduces chondrocyte apoptosis, attenuates oxidative stress by upregulating the NRF2/HO-1 antioxidant pathway, and alleviates cartilage damage and bone proliferation. However, major revision is needed.

Major:

1.The manuscript mentions that 3-week-old mouse chondrocytes were used, and these cells are in a proliferative state, which may not fully represent the condition of adult chondrocytes. Since KOA mainly affects adults, verifying the effect of APT on adult mouse chondrocytes would increase the clinical relevance of the study.

2.The manuscript also states that the study focused only on cartilage destruction and changes in subchondral bone, without addressing other tissues like the infrapatellar fat pad and synovium, which also play important roles in the pathogenesis of KOA. If APT can exert protective effects on these related tissues, particularly by reducing synovial inflammation or mitigating iron overload damage in synovial cells, it would provide stronger evidence for APT as a broad-spectrum candidate drug for KOA treatment.

3.Current research has only evaluated the short-term effects of APT at specific concentrations (5-10 μM). Further investigation is needed to study the long-term therapeutic effects of APT and whether an optimal dosage window exists.

4.The author focuses on the impact of APT on the NRF2-HO-1 pathway but does not explore whether it regulates iron metabolism-related genes (such as FPN, DMT1, and TfR1) to reduce iron overload-induced cellular damage. If APT can regulate iron metabolism pathways, decreasing iron accumulation or enhancing iron clearance, it would provide a novel mechanism for treating iron overload diseases and help explain its multiple antioxidant and anti-apoptotic effects.

Minor:

1.In the introduction, the authors mention, "Iron overload is broadly recognized as an essential factor in the injury." The background could be more comprehensive. References such as PMID: 33080340 and PMID: 37875229 could be cited.

2.This manuscript contains numerous grammar and spelling errors, as well as areas for improvement. Line 379: The authors could remove "iron overload alone may not be sufficient to induce KOA in animal studies" as this sentence is repeated later.

3.Line 385: Remove the unnecessary "of" in "Approximately 5–10 μM of APT significantly counteracted."

4.Line 390: Use "severely restricted and is only activated" instead of "severely restrict and only activate."

5.Line 394: "APT effectively reversed the effect of iron overload by promoting" should be rephrased to avoid repetition of "effect."

6.Line 403: "Apoptosis may be induced by this damage" could be rephrased for better clarity as: "This damage may induce apoptosis."

7.Line 410: Remove the second "HO" in "As an inducible 32-kDa protein" for clarity.

8.Line 424: For clarity, rephrase "The decrease in NRF2 expression" to: "The reduction in NRF2 expression."

6. PLOS authors have the option to publish the peer review history of their article (what does this mean?). If published, this will include your full peer review and any attached files.

Reviewer #1: **Yes: **Chen Ling

---

## [Author Response · Author response to Decision Letter 1]

31 Dec 2024

1.The manuscript mentions that 3-week-old mouse chondrocytes were used, and these cells are in a proliferative state, which may not fully represent the condition of adult chondrocytes. Since KOA mainly affects adults, verifying the effect of APT on adult mouse chondrocytes would increase the clinical relevance of the study.

Response: Thank you for your suggestion. Chondrocytes from young mice were chosen as an in vitro model due to the superior proliferative capacity of such chondrocytes, and there are more examples of chondrocytes extracted using younger week-old mice (e.g. DOI: 10.1002/adhm.202300315), but this does diminish the significance of this study.

We therefore extracted chondrocytes from the joints of adult mice (20week), but these cells are slow-growing and spontaneously apoptotic at P3, so we used P2 for Alisin Blue staining as well as apoptosis detection. The results showed that APT was able to alleviate collagen production due to iron overload in a concentration-dependent manner and reduce the rate of apoptosis in iron overload, which we added in Supplementary Figure 1, which you can access in the new manuscript.

2.The manuscript also states that the study focused only on cartilage destruction and changes in subchondral bone, without addressing other tissues like the infrapatellar fat pad and synovium, which also play important roles in the pathogenesis of KOA. If APT can exert protective effects on these related tissues, particularly by reducing synovial inflammation or mitigating iron overload damage in synovial cells, it would provide stronger evidence for APT as a broad-spectrum candidate drug for KOA treatment.

Response: Thank you for your suggestion that the synovium indeed has an important role in arthritis, as recently elucidated by (DOI: 10.1126/scittranslmed.adf4590).Based on your suggestion we designed two protocols, on the one hand, the synovium contains mainly synovial fibroblasts as well as macrophages, fibroblasts are very important in rheumatoid arthritis, while they are relatively less studied in osteoarthritis or iron overload arthritis. Whereas macrophages are key cells for the release of inflammatory mediators in osteoarthritis, since we used mice as an animal model and we lacked experience with the extraction of mouse synovial macrophages in our in vitro studies, we used RAW264.7 as a cellular model and investigated whether there was a change in the release of IL-1β and TNF-α in the supernatant of the culture medium after in vitro FAC intervention with RAW264.7.Unfortunately the release of these classical inflammatory mediators did not positively correlate with iron overload.

In another line of thought we performed H&E staining of the sections to visualize the synovial and adipose tissue located under the patella, anterior to the knee, and found that the DMM and ID+DMM groups had significant synovial hyperplasia compared to the CTRL group, and it was more pronounced in the ID+DMM group, and that the APT could reverse this trend.We integrated these results with CT reconstructions and you can see them in the new manuscript.

Taken together, we provide some limited evidence that APT attenuates synovial changes in iron overload arthritis, but it is not certain whether this phenomenon results from direct targeting of iron overload injury in the synovium or from attenuating the inflammatory milieu by alleviating cartilage changes.After all, iron overload alone in vitro does not induce macrophages to secrete inflammatory factors. Therefore this question needs to be validated by a thoughtful experimental design in a new program, which is our next goal. Further discussion is difficult in our current manuscript.

3.Current research has only evaluated the short-term effects of APT at specific concentrations (5-10 μM). Further investigation is needed to study the long-term therapeutic effects of APT and whether an optimal dosage window exists.

Response: Thank you for your suggestion.We designed this study using a commonly used preliminary drug study protocol, and most published studies of this type use intervention protocols of 8-10 weeks in length; however, this protocol does not allow for assessment of long-term effects, and we apologize for the inconvenience this has caused you. We will follow up with a longer-term toxicology and efficacy evaluation of the drug, but due to the length of the study, this will not be included in this manuscript.

4.The author focuses on the impact of APT on the NRF2-HO-1 pathway but does not explore whether it regulates iron metabolism-related genes (such as FPN, DMT1, and TfR1) to reduce iron overload-induced cellular damage. If APT can regulate iron metabolism pathways, decreasing iron accumulation or enhancing iron clearance, it would provide a novel mechanism for treating iron overload diseases and help explain its multiple antioxidant and anti-apoptotic effects.

Response: Thank you for your suggestion.Your idea has inspired us to complement our testing of changes in the metabolic genes FPN, DMT1 and TfR1 in our cellular model, with very interesting results. First, iron overload significantly altered these genes. Specifically, DMT1 and FPN decreased in iron overload, while TfR1 increased. Of these three genes, FPN, who plays a role in the exocytosis of iron ions, did not change significantly after APT intervention, but TfR1 was regulated by APT intervention. Since FPN, DMT1 and TfR1 all regulate iron metabolism, but the regulatory relationship reflected in PCR is not very certain. We therefore decided to use calcium xanthophyll to detect unstable intracellular iron pools, which was used to determine whether APT altered intracellular iron accumulation, and found that APT did not have a significant iron uptake or efflux effect. This part of the results we put in Supplementary Figure 2 and you can see this in the latest manuscript.

5. In the introduction, the authors mention, "Iron overload is broadly recognized as an essential factor in the injury." The background could be more comprehensive. References such as PMID: 33080340 and PMID: 37875229 could be cited.

Response: Thank you for the reminder. Citation has been added as per your suggestion.

6.This manuscript contains numerous grammar and spelling errors, as well as areas for improvement. Line 379: The authors could remove "iron overload alone may not be sufficient to induce KOA in animal studies" as this sentence is repeated later.

Response: Thank you for the reminder. This part of the text has been modified in accordance with your suggestions.

7. Line 385: Remove the unnecessary "of" in "Approximately 5–10 μM of APT significantly counteracted."

Response: Thank you for the reminder. This part of the text has been modified in accordance with your suggestions.

8. Line 390: Use "severely restricted and is only activated" instead of "severely restrict and only activate."

Response: Thank you for the reminder. This part of the text has been modified in accordance with your suggestions.

9.Line 394: "APT effectively reversed the effect of iron overload by promoting" should be rephrased to avoid repetition of "effect."

Response: Thank you for the reminder. This part of the text has been modified in accordance with your suggestions.

10.Line 403: "Apoptosis may be induced by this damage" could be rephrased for better clarity as: "This damage may induce apoptosis."

Response: Thank you for the reminder. This part of the text has been modified in accordance with your suggestions.

11.Line 410: Remove the second "HO" in "As an inducible 32-kDa protein" for clarity.

Response: Thank you for the reminder. This part of the text has been modified in accordance with your suggestions.

12.Line 424: For clarity, rephrase "The decrease in NRF2 expression" to: "The reduction in NRF2 expression."

Response: Thank you for the reminder. This part of the text has been modified in accordance with your suggestions.

13.To comply with PLOS ONE submissions requirements, in your Methods section, please provide additional information regarding the experiments involving animals and ensure you have included details on (1) methods of sacrifice, (2) methods of anesthesia and/or analgesia, and (3) efforts to alleviate suffering.

Response: Thank you for the reminder. We have added the required content to the animal testing section.

14.We noticed you have some minor occurrence of overlapping text with the following previous publication(s), which needs to be addressed.

Response: Thank you for the reminder. We've fixed the part with duplicates.

15.Please state what role the funders took in the study.

Response: Thank you for the reminder. We've stated what role the funders took in the study.

---

## [Decision Letter · Decision Letter 1]

Alpinetin protects against iron overload related osteoarthritis via NRF2/HO-1 pathway

PONE-D-24-39558R1

Dear Dr. Zhou,

We’re pleased to inform you that your manuscript has been judged scientifically suitable for publication and will be formally accepted for publication once it meets all outstanding technical requirements.

Kind regards,

Xindie Zhou

Academic Editor

PLOS ONE

Additional Editor Comments (optional):

Reviewers' comments:

Reviewer's Responses to Questions

**Comments to the Author**

1. If the authors have adequately addressed your comments raised in a previous round of review and you feel that this manuscript is now acceptable for publication, you may indicate that here to bypass the “Comments to the Author” section, enter your conflict of interest statement in the “Confidential to Editor” section, and submit your "Accept" recommendation.

Reviewer #1: All comments have been addressed

2. Is the manuscript technically sound, and do the data support the conclusions?

Reviewer #1: Yes

3. Has the statistical analysis been performed appropriately and rigorously? 

Reviewer #1: Yes

4. Have the authors made all data underlying the findings in their manuscript fully available?

Reviewer #1: Yes

5. Is the manuscript presented in an intelligible fashion and written in standard English?

Reviewer #1: Yes

6. Review Comments to the Author

Reviewer #1: (No Response)

7. PLOS authors have the option to publish the peer review history of their article (what does this mean?). If published, this will include your full peer review and any attached files.

Reviewer #1: No

---

## [Editor Report · Acceptance letter]

PONE-D-24-39558R1

PLOS ONE

Dear Dr. Zhou,

I'm pleased to inform you that your manuscript has been deemed suitable for publication in PLOS ONE. Congratulations! Your manuscript is now being handed over to our production team.

Kind regards,

on behalf of

Dr. Xindie Zhou

Academic Editor

PLOS ONE